# Sample-efficient decoding of visual stimuli from fMRI through inter-individual functional alignment

**Alexis Thual** *  *alexis.thual@inria.fr*
*Cognitive Neuroimaging Unit, INSERM, CEA, CNRS, NeuroSpin center, Gif sur Yvette, France*
*Mind, Inria Paris-Saclay, Palaiseau, France*
*Inserm, Collège de France, Paris, France*

**Yohann Benchetrit**
*Meta AI*

**Félix Geilert**
*Meta AI*

**Jérémy Rapin**
*Meta AI*

**Iurii Makarov**
*Meta AI*

**Stanislas Dehaene**
*Cognitive Neuroimaging Unit, INSERM, CEA, CNRS, NeuroSpin center, Gif sur Yvette, France*
*Inserm, Collège de France, Paris, France*

**Bertrand Thirion**
*Mind, Inria Paris-Saclay, Palaiseau, France*

**Hubert Banville**
*Meta AI*

**Jean-Rémi King**  *jeanremi@meta.com*
*Meta AI*
*Laboratoire des systèmes perceptifs, École normale supérieure*
*PSL University*

**Reviewed on OpenReview:** *https://openreview.net/forum?id=qvJraN5ODT*

---

*Work done while interning at Meta AI

## Abstract

Deep learning is leading to major advances in the realm of brain decoding from functional Magnetic Resonance Imaging (fMRI). However, the large inter-individual variability in brain characteristics has constrained most studies to train models on one participant at a time. This limitation hampers the training of deep learning models, which typically requires very large datasets. Here, we propose to boost brain decoding of videos and static images across participants by aligning brain responses of training and left-out participants. Evaluated on a retrieval task, compared to the anatomically-aligned baseline, our method halves the median rank in out-of-subject setups in low-data regimes. It also outperforms classical within-subject approaches when fewer than 100 minutes of data is available for the tested participant. Furthermore, we show that our alignment framework handles multiple subjects, which improves accuracy upon classical single-subject approaches. Finally, we show that this method aligns neural representations in accordance with brain anatomy. Overall, this study lays the foundations for leveraging extensive neuroimaging datasets and enhancing the decoding of individual brains when a limited amount of brain-imaging data is available.

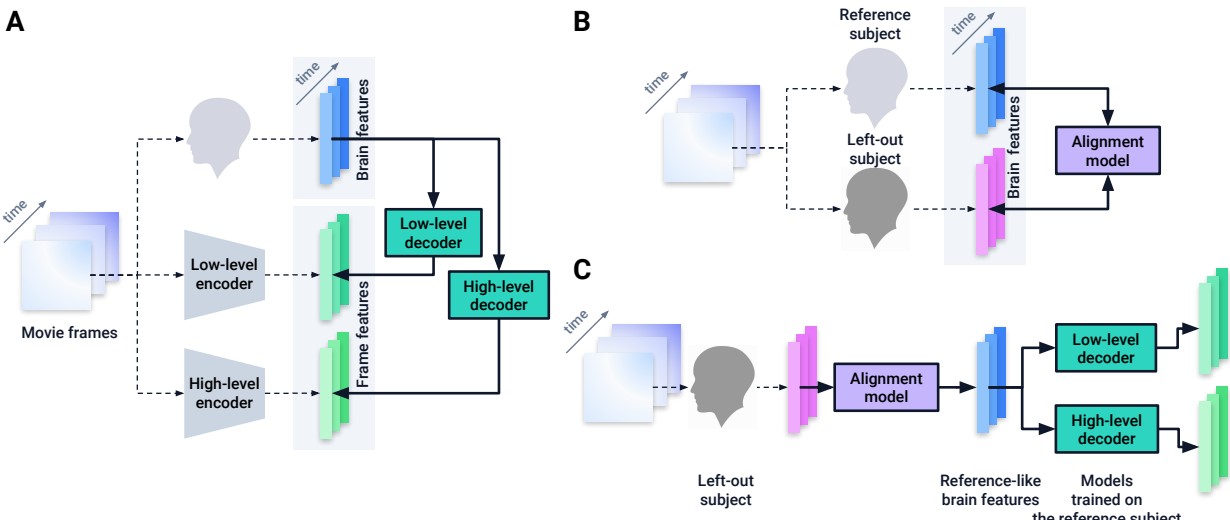

Figure 1: **General outline of video decoding from BOLD fMRI signal in left-out participants A.** For each frame associated with a brain volume, one computes its low- and high-level latent representations using pre-trained encoders. Then, brain decoders (green) can then be fitted to map brain features onto each of these latent representations. **B.** The BOLD signal acquired from two participants watching the same movie can be used to derive an alignment model (purple) that maps voxels from the two participants based on functional similarity. **C.** Then, this alignment model can be used to transform brain features of the left-out participant into the brain features that match those of the reference participant. In particular, this allows one to use decoders that have been trained on a lot of data coming from a reference participant, and apply them on a left-out participant for whom less data was collected.

## 1 Introduction

**Decoding brain activity** The generative capabilities of deep learning have recently unlocked decoding mental representations from brain activity. Originally restricted to linear models (Mitchell et al., 2004; Harrison & Tong, 2009; Haynes & Rees, 2006), the decoding of brain activity can now be carried out with deep learning techniques. In particular, using functional Magnetic Resonance Imaging (fMRI) signals, significant progress has been made in the decoding of images (Ozcelik & VanRullen, 2023; Chen et al., 2023a;

Scotti et al., 2023; Takagi & Nishimoto, 2023; Gu et al., 2023; Ferrante et al., 2023; Mai & Zhang, 2023), speech (Tang et al., 2023), and videos (Kupershmidt et al., 2022; Wen et al., 2018; Wang et al., 2022; Chen et al., 2023b; Lahner et al., 2023; Phillips et al., 2022).

**The bottleneck of inter-subject variability**  A core issue is that brain organization is highly variable across participants, which makes it challenging to train a single model on multiple participants using fMRI data. Therefore, with few noteworthy exceptions (Haxby et al., 2020; Ho et al., 2023), studies typically train a brain decoder on a single participant at a time. With this constraint in mind, major effort has been put towards building fMRI datasets collecting a lot of data in a limited number of participants (Allen et al., 2022; Wen et al., 2017; LeBel et al., 2023; Pinho et al., 2018). Nonetheless, the necessity to train and test models on a single participant constitutes a major impediment to using notoriously data-hungry deep learning approaches. In addition, generalization to new individuals is essential to the validation of discoveries.

**Functional alignment**  Several methods can align the functional organization – on top of the anatomy – of multiple brains, and thus offer a potential solution to inter-individual variability: differentiable warps of the cortical surface (Robinson et al., 2014), rotations between brain voxels in the functional space (Haxby et al., 2011), shared response models (Chen et al., 2015; Richard et al., 2020), permutations of voxels minimizing an optimal transport cost (Bazeille et al., 2019; Thual et al., 2022), or combinations of these approaches (Feilong et al., 2022). More recently, several studies rely on deep learning models trained in a self-supervised fashion to build an embedding of brain activity, in hope that it could be meaningful across participants (Thomas et al., 2022; Chen et al., 2023a). However, to this day, it is not clear which of these methods offers the best performance and generalization capabilities (Bazeille et al., 2021).

**Approach**  It is currently unknown whether any of the aforementioned methods improve the decoding of naturalistic stimuli such as videos, and how such hypothetical gain would vary with the amount of fMRI recording available in a given a participant. To address this issue, we leverage fMRI recordings of training participants to boost the decoding of videos and static images in a single left-out participant, as illustrated in Figure 1. This requires fitting two models: an alignment model and a brain decoder. The alignment aims at making brain responses of a left-out participant most similar to those of a reference participant. Here, we leverage optimal transport to compute this transformation using functional and anatomical data from both participants. The brain decoder – which we will refer to as the *decoder* – consists of a linear regression trained to predict the latent representations of movie frames or static images from the corresponding BOLD signals or beta coefficients. We evaluate video and image decoding in different setups. In particular, we assess (1) whether decoders generalize to participants on which they were not trained, (2) whether training a decoder on data from multiple participants improves performance and (3) the extent to which functional alignment improves the aforementioned setups.

**Contributions**  We first confirm the feasibility of decoding, from 3 Tesla (3T) fMRI, the semantics of videos watched by the participants (Wen et al., 2017). We verify that this approach also performs well for the decoding of static images from 7T fMRI data (Allen et al., 2022). Our study makes three main novel contributions:

1. Compared to the baseline, functional alignment across participants boosts visual semantics decoding performance in left-out participants when the latter have a limited amount of data

2. Training a decoder on multiple functionally aligned participants yields a model with improved performance compared to training one model per participant, but anatomical alignment does not

3. The resulting alignments, computed from movie-watching data, are anatomically coherent

## 2 Methods

Our goal is to decode visual stimuli seen by individuals from their brain activity. To this end, we train a linear model to predict latent representations – shortened as *latents* – of these visual stimuli from BOLD fMRI signals recorded in participants watching naturalistic videos.

In the data under study, brains are typically imaged at a rate of one scan every 2 seconds. During this period, a participant sees 60 video frames on average, or a static image for the case of Allen et al. (2022). For simplicity, we consider the restricted problem of decoding only the first video frame seen by participants at each brain scan. Formally, regardless of the dataset, for a given participant, let $\boldsymbol{X} \in \mathbb{R}^{n,v}$ be the BOLD response collected in $v$ voxels over $n$ brain scans and $\boldsymbol{Y} \in \mathbb{R}^{n,m}$ the $m$-dimensional latent representation of each selected video frame.

### 2.1 Decoding

**Brain input** There is a time *lag* between the moment a stimulus is played and the moment it elicits a maximal BOLD response in the brain (Glover, 1999). Moreover, the effect induced by this stimulus might span over multiple consecutive brain volumes. To account for these effects, we use a standard Finite Impulse Response (FIR) approach. It consists in fitting the decoder on a time-lagged, multi-volume version of the BOLD response. In particular, we refer to the number of brain volumes to aggregate together in the FIR approach as the *window size*. Different *aggregation functions* can be used, such as stacking or averaging. Figure S2 describes these concepts visually.

**Video output** The matrix of latent features $\boldsymbol{Y}$ is obtained by using a pre-trained image encoder on each video frame and concatenating all obtained vectors in $\boldsymbol{Y}$. Similarly to Ozcelik & VanRullen (2023), and as illustrated in Figure 1.A, we seek to predict CLIP $257 \times 768$ (high-level) and VD-VAE (low-level) latent representations . We use visual – as opposed to textual – CLIP representations (Radford et al., 2021). For comparison, we reproduce our approach on latent representations from CLIP CLS (high-level) and AutoKL (low-level), which happen to be much smaller [1] and are computationally easier to fit.

**Model** Fitting the decoder consists in deriving $\boldsymbol{W} \in \mathbb{R}^{v,m}$, $\boldsymbol{b} \in \mathbb{R}^m$ the solution of a Ridge regression problem – i.e. a linear regression with L2 regularization – predicting $\boldsymbol{Y}$ from $\boldsymbol{X}$. Note that $v$, the number of vertices, is the same for the brain decoder and for the brain alignment module.

**Evaluation** We evaluate the performance of the decoder with retrieval metrics. Let us denote $\boldsymbol{X}_{\text{train}}$ and $\boldsymbol{Y}_{\text{train}}$ the brain and latent features used to train the decoder, $\boldsymbol{X}_{\text{test}}$ and $\boldsymbol{Y}_{\text{test}}$ those to test the decoder, and $\hat{\boldsymbol{Y}} \triangleq \boldsymbol{W}\boldsymbol{X}_{\text{test}} + \boldsymbol{b}$ the predicted latents. We ensure that the train and test data are disjoint.

We randomly draw a retrieval set $K$ of 499 frames without replacement from the test data. For each pair $(\hat{\boldsymbol{y}}, \boldsymbol{y})$ of predicted and ground truth latents, one derives their cosine similarity score $s(\hat{\boldsymbol{y}}, \boldsymbol{y})$, as well as similarity scores to all latents $\boldsymbol{y}_{\text{neg}}$ of the retrieval set $s(\hat{\boldsymbol{y}}, \boldsymbol{y}_{\text{neg}})$. Let us denote $r_K(\hat{\boldsymbol{y}}, \boldsymbol{y})$ the rank of $\boldsymbol{y}$, which we define as the number of elements of $K$ whose similarity score to $\hat{\boldsymbol{y}}$ is larger than $s(\hat{\boldsymbol{y}}, \boldsymbol{y})$. In order for the rank to not depend on the size of $K$, we define the *relative rank* as $r(\hat{\boldsymbol{y}}, \boldsymbol{y})/|K|$. Finally, one derives the median relative rank $\text{MR}(\hat{\boldsymbol{Y}}, K)$:

$$r_K(\hat{\boldsymbol{y}}, \boldsymbol{y}) \triangleq \left| \left\{ \boldsymbol{y}_{\text{neg}} \in K \mid s(\hat{\boldsymbol{y}}, \boldsymbol{y}_{\text{neg}}) > s(\hat{\boldsymbol{y}}, \boldsymbol{y}) \right\} \right|$$

$$\text{MR}(\hat{\boldsymbol{Y}}, K) \triangleq \text{median}\left( \left\{ r_K(\hat{\boldsymbol{y}}, \boldsymbol{y})/|K| \,, \forall \, (\hat{\boldsymbol{y}}, \boldsymbol{y}) \right\} \right)$$

### 2.2 Brain alignment

**Anatomical alignment** As a baseline, we consider the alignment method implemented in Freesurfer (Fischl, 2012), which relies on anatomical information to project each participant onto a surface template

---

[1] Dimensions for CLIP CLS: 768 ; CLIP $257 \times 768$ : $257 \times 768 = 197\,376$ ; AutoKL: $4 \times 32 \times 32 = 4\,096$ ; VD-VAE: $2 \times 2^4 + 4 \times 2^8 + 8 \times 2^{10} + 16 \times 2^{12} + 2^{14} = 91\,168$

of the cortex (in our case *fsaverage5*). Consequently, brain data from all participants lie on a mesh of size $v = 10\,242$ vertices per hemisphere.

**Functional alignment**   On top of the aforementioned anatomical alignment, we apply a recent method from Thual et al. (2022) denoted as Fused Unbalanced Gromov-Wasserstein (FUGW) [2]. As illustrated in Figure 1.B, this method consists in using functional data to train an alignment that transforms brain responses of a given left-out participant into the brain responses of a reference participant. This approach can be seen as a soft permutation of voxels [3] of the left-out participant which maximizes the functional similarity to voxels of the reference participant.

Formally, for a left-out participant, let $\boldsymbol{D}^{\text{out}} \in \mathbb{R}^{v,v}$ be the matrix of anatomical distances between vertices on the cortex, and $\boldsymbol{w}^{\text{out}} \in \mathbb{R}_+^v$ a probability distribution on vertices. $\boldsymbol{w}^{\text{out}}$ can be interpreted as the relative importance of vertices; without prior knowledge, we use the uniform distribution. Reciprocally, we define $\boldsymbol{D}^{\text{ref}}$ and $\boldsymbol{w}^{\text{ref}}$ for a reference participant. Note that, in the general case, $v$ can be different from one participant to the other, although we simplify notations here.

We derive a transport plan $\boldsymbol{P} \in \mathbb{R}^{v,v}$ to match the vertices of the two participants based on functional similarity, while preserving anatomical organisation. For this, we simultaneously optimize multiple constraints, formulated in the loss function $\mathcal{L}_\Theta(\boldsymbol{P})$ described in Equation 1:

$$
\begin{array}{c}
\overset{\text{Wasserstein loss}}{} \qquad \overset{\text{Gromov Wasserstein loss}}{} \\
\mathcal{L}_\Theta(\boldsymbol{P}) \triangleq (1-\alpha)\ \boxed{\mathcal{L}_{\text{W}}(\boldsymbol{P})} + \alpha\ \boxed{\mathcal{L}_{\text{GW}}(\boldsymbol{P})} \\
+ \rho\ \boxed{\mathcal{L}_{\text{U}}(\boldsymbol{P})} + \varepsilon\ \boxed{H(\boldsymbol{P})} \\
\underset{\text{Marginal constraints}}{} \qquad \underset{\text{Regularization}}{}
\end{array}
\tag{1}
$$

Each component of the loss is expressed as follows:

- $\mathcal{L}_{\text{W}}(\boldsymbol{P}) \triangleq \displaystyle\sum_{0 \le i,j < v} ||\boldsymbol{X}_i^{\text{out}} - \boldsymbol{X}_j^{\text{ref}}||_2^2\ \boldsymbol{P}_{i,j}$

- $\mathcal{L}_{\text{GW}}(\boldsymbol{P}) \triangleq \displaystyle\sum_{0 \le i,k,j,l < v} |\boldsymbol{D}_{i,k}^{\text{out}} - \boldsymbol{D}_{j,l}^{\text{ref}}|^2\ \boldsymbol{P}_{i,j}\ \boldsymbol{P}_{k,l}$

- $\mathcal{L}_{\text{U}}(\boldsymbol{P}) \triangleq \text{KL}(\boldsymbol{P}_{\#1} \otimes \boldsymbol{P}_{\#1} | \boldsymbol{w}^{\boldsymbol{s}} \otimes \boldsymbol{w}^{\boldsymbol{s}}) + \text{KL}(\boldsymbol{P}_{\#2} \otimes \boldsymbol{P}_{\#2} | \boldsymbol{w}^{\boldsymbol{t}} \otimes \boldsymbol{w}^{\boldsymbol{t}})$

- $H(\boldsymbol{P}) \triangleq \text{KL}\big(\boldsymbol{P} \otimes \boldsymbol{P} \mid (\boldsymbol{w}^{\boldsymbol{s}} \otimes \boldsymbol{w}^{\boldsymbol{t}}) \otimes (\boldsymbol{w}^{\boldsymbol{s}} \otimes \boldsymbol{w}^{\boldsymbol{t}})\big)$

Here, $\text{KL}(\cdot, \cdot)$ denotes the Kullback-Leibler divergence, $\boldsymbol{P}_{\#1} \triangleq (\sum_j \boldsymbol{P}_{i,j})_{0 \le i < n}$ is the first marginal of $\boldsymbol{P}$, $\boldsymbol{P}_{\#2} \triangleq (\sum_i \boldsymbol{P}_{i,j})_{0 \le j < n}$ is the second marginal of $\boldsymbol{P}$, $\alpha \in [0,1]$, $\rho \in \mathbb{R}_+$ are the hyper-parameters setting the relative importance of each constraint, and $\Theta \triangleq (\boldsymbol{X}^{\text{out}}, \boldsymbol{X}^{\text{ref}}, \boldsymbol{D}^{\text{out}}, \boldsymbol{D}^{\text{ref}}, \alpha, \rho, \varepsilon)$.

Following Thual et al. (2022), we minimize $\mathcal{L}_\Theta(\boldsymbol{P})$ with 10 iterations of a block coordinate descent algorithm (Séjourné et al., 2021), each running $1\,000$ Sinkhorn iterations (Cuturi, 2013). Subsequently, we define $\phi_{\text{out}\to\text{ref}}\colon \boldsymbol{X} \mapsto (\boldsymbol{P}^T \boldsymbol{X}^T) \oslash \boldsymbol{P}_{\#2} \in \mathbb{R}^{n,v}$ where $\oslash$ is the element-wise division, a function which transports any matrix of brain features from the left-out participant to the reference participant. To simplify notations, for any $\boldsymbol{X}$ defined on the left-out participant, we define $\boldsymbol{X}^{\text{out}\to\text{ref}} \triangleq \phi_{\text{out}\to\text{ref}}(\boldsymbol{X})$.

**Hyper-parameters selection for functional alignment**   We use default parameters shipped with version 0.1.0 of FUGW. Namely, $\alpha$, which controls the balance between Wasserstein and Gromov-Wasserstein losses – i.e. how important functional data is compared to anatomical data – is set to 0.5. Empirically, we see that $\alpha = 0.5$ yields values for the Wasserstein loss which are larger than that of the Gromov-Wasserstein loss, meaning that functional data drives these alignments. Secondly, $\rho$, which sets the importance of marginal

---

[2]https://alexisthual.github.io/fugw
[3]We use the words *voxel* (volumetric pixel) or *vertex* (point on a mesh) indifferently.

constraints – i.e. to what extent more or less mass can be transported to / from each voxel – is set to 1. Empirically, this value leads to all voxels being transported / matched with equal importance. Finally, $\varepsilon$, which controls for entropic regularization – i.e. how blurry computed alignments will be – is set to $10^{-4}$. Empirically, this value yields alignments which are anatomically very sharp, i.e source voxels are matched with a handful of target voxels only (and vice-versa).

## 3 Experimental setup

### 3.1 Decoding and alignment setups

Our main methodological contribution consists in training and evaluating brain decoders in a variety of setups in which participants have been functionally aligned or not.

**Within- *vs* out-of-subject**  Let us consider a decoder trained on data $(\boldsymbol{X}_{\text{train}}^{S_1}, \boldsymbol{Y}_{\text{train}}^{S_1})$ from a given participant. The *within-subject* setup consists in testing it on left-out data $(\boldsymbol{X}_{\text{test}}^{S_1}, \boldsymbol{Y}_{\text{test}}^{S_1})$ acquired in the same participant. The *out-of-subject* setup consists in testing it on data $(\boldsymbol{X}_{\text{test}}^{S_2}, \boldsymbol{Y}_{\text{test}}^{S_2})$ acquired in a left-out participant.

**Single- *vs* multi-subject**  The *single-subject* setup consists in training a decoder using data from one participant only. The *multi-subject* setup consists in training a decoder using data from multiple participants. In this study, data from several participants are stacked, resulting in a matrix $\boldsymbol{X}_{\text{multi}} \in \mathbb{R}^{n_1 + \dots + n_p, v}$ and $\boldsymbol{Y}_{\text{multi}} \in \mathbb{R}^{n_1 + \dots + n_p, m}$ , where $p$ is the number of participants.

**Aligned *vs* un-aligned**  Let $S_1$ be the *reference* participant. In the out-of-subject and multi-subject setups, data coming from different participants can be *functionally aligned* – or not – to that of the *reference* participant. It modifies these respective setups as follows: (1) in the out-of-subject case, it corresponds to aligning $S_2$ onto $S_1$, such that a decoder trained on $S_1$ will be tested on $\boldsymbol{X}_{\text{test}}^{S_2 \to S_1}$, $\boldsymbol{Y}_{\text{test}}^{S_2}$, (2) in the multi-subject case, all participants are aligned to $S_1$ and the decoder is trained on a concatenation of $\boldsymbol{X}^{S_1}, \boldsymbol{X}^{S_2 \to S_1}, ..., \boldsymbol{X}^{S_p \to S_1}$ (see notations introduced at the end of section 2.2) and $\boldsymbol{Y}^{S_1}, ..., \boldsymbol{Y}^{S_p}$.

Setups of interest are visually described in Figure 3.A.

**Evaluation under different data regimes**  Note that the alignment and decoding models do not need to be fitted using the same amount of data. In particular, we are interested in evaluating out-of-subject performance in setups where a lot of data is available for the reference participant, and little data is available for the left-out participant: this would typically be the case in clinical setups where, usually, little data is available in patients. In this case, we evaluate whether it is possible to use this small amount of data to align the left-out participant onto the reference participant, and have the left-out participant benefit from a decoder previously trained on a lot of data.

### 3.2 Datasets

We analyze two fMRI datasets. The first dataset (Wen et al., 2017) comprises 3 human participants who watched 688 minutes of video. The videos consists of 18 train segments of 8 minutes each and 5 test segments of 8 minutes each. Each training segment was presented twice. Each test segment was presented 10 times. Each segment consists of a sequence of roughly 10-second video clips. The fMRI data was acquired at 3T, 3.5mm isotropic spatial resolution and 2-second temporal resolution. It was minimally pre-processed with the same pre-processing pipeline as that of the Human Connectome Project (Glasser et al., 2013). In particular, data from each participant are projected onto a common volumetric anatomical template. Similarly to prior work on this dataset (Wen et al., 2018; Kupershmidt et al., 2022; Wang et al., 2022), we use runs related to the first 18 video segments - 288 minutes - as training data, and runs related to the last 5 video segments as test data. Thus, it amounts to 8640 training samples and 1200 test samples per individual.
The second dataset (Allen et al., 2021) – denoted as the Natural Scenes Dataset (NSD) – comprises 8 participants who are shown 10 000 static images three times. They were scanned over 40 sessions of 60

minutes, amounting to 2 400 minutes of data. This amounts to 72 000 samples per individual. Instead of raw BOLD signal, we leverage precomputed per-trial regression coefficients accessible online. See supplementary section A.6 for more details.

### 3.3 Preprocessing

For the Wen et al. (2017) dataset, we implement minimal additional preprocessing steps for each participant separately. For this, we (1) project all volumetric data onto the FreeSurfer average surface template *fsaverage5* (Fischl, 2012), then (2) regress out cosine drifts in each vertex and each run and finally (3) center and scale each vertex time-course in each run. Figure S1 gives a visual explanation as to why the last two steps are needed. The first two steps are implemented with Nilearn (Abraham et al., 2014) [4] and the last one with Scikit-Learn (Pedregosa et al., 2011).

Additionally, for a given participant, we try out two different setups: a first one where runs showing the same video are averaged, and a second one where they are stacked.

The Allen et al. (2021) dataset is already preprocessed by the original authors, and maps of beta coefficients from a General Linear Model are accessible online.

### 3.4 Hyper-parameters selection for decoders

To train decoders, we use the same regularization coefficient $\alpha_{\mathrm{ridge}}$ across latent types and choose it by running a cross-validated grid search on folds of the training data. We find that results are robust to using different values and therefore set $\alpha_{\mathrm{ridge}} = 50\,000$. Similarly, values for lag, window size and aggregation function are determined through a cross-validated grid search. The loss used to determine these parameters in the relative median rank introduced in 2.1.

## 4 Results

### 4.1 Within-subject prediction of visual representations from BOLD signal and retrieval of visual inputs

Table 1: **Within-subject metrics for all participants and all latent types on the test set** Reported metrics are relative median rank ↓ (MR) of retrieval on a set of 500 samples, top-5 accuracy % ↑ (Acc) of retrieval on a set of 500 samples. Chance level is at 50.0 and 1.0 for these two metrics respectively. These results were averaged across 50 retrieval sets, hence results are reported with a standard error of the mean (SEM) smaller than 0.01. The *Dummy* (D) model systematically predicts the mean latent representation of the training set and achieves chance level.

|  | CLIP 257 × 768 | | VD-VAE | | CLIP CLS | | AutoKL | |
|---|---|---|---|---|---|---|---|---|
|  | MR | Acc | MR | Acc | MR | Acc | MR | Acc |
| Dummy | 50.0 | 1.0 | 50.0 | 1.0 | 50.0 | 1.0 | 50.0 | 1.0 |
| S1 | 9.4 | 13.8 | 29.9 | 3.0 | 15.1 | 8.4 | 24.9 | 3.9 |
| S2 | 6.8 | 16.4 | 30.2 | 3.5 | 10.6 | 10.5 | 21.8 | 3.8 |
| S3 | 7.8 | 13.6 | 28.5 | 3.1 | 11.0 | 9.9 | 26.0 | 3.3 |

We report video decoding results on the retrieval task in Table 1. For all three participants of the Wen 2017 dataset, and for all four types of latent representations considered, a Ridge regression fitted within-subject achieves significantly above-chance performance. Besides, performance varies across participants, although well-performing participants reach good performance on all types of latents.

Results reported in Table 1 were obtained for a lag of 2 brain volumes (i.e. 4 seconds since TR = 2 seconds) and a window size of 2 brain volumes that were averaged together (see definitions in section 2.1). These parameters were chosen after running a k-fold cross-validated grid search for lag values ranging from 1 to

---

[4] https://nilearn.github.io

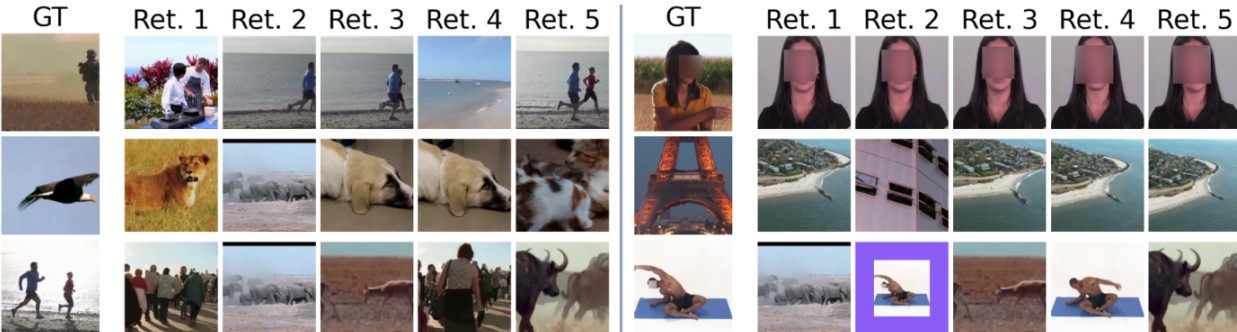

Figure 2: **Image retrievals using predicted latent representations of CLIP 257 × 768 latents** We use a model fitted on Subject 2 (S2) and predict the latent representation of unseen videos (test set). Ground truth (GT) images featured within the first 5 retrieved (Ret.) images are indicated with a bold purple border. In a given row, images which appear similar across columns are actually different frames of the same video clip. Images featuring human faces were blurred. More cases are available in supplementary Figure S4.

5, a window size ranging from 1 to 3, and 2 possible aggregation functions for brain volumes belonging to the same window (namely averaging and stacking). Figure S3 shows results using the averaging aggregation function for different values of lag and window size, averaged across participants. These results were obtained by stacking all runs of the training dataset, as opposed to averaging repetitions of the same video clip. The two approaches yielded very similar metrics. We give more details in section 4.3.

Figure 2 shows retrieved images for Subject 2. Qualitatively, we observe that retrieved images often fit the theme of images shown to participants (with categories like indoor sports, human faces, animals, etc.), yet with occasional failures.

## 4.2 Out-of-subject decoding and multi-subject training

As illustrated in Figure 3, models trained on one participant do not generalise well to other participants: using CLIP 257 × 768, the within-subject and out-of-subject median rank (MR) are respectively 8.0 and 17.2 on average. However, functional alignment allows to reduce the median rank back to 11.1 on average. In particular, we show that left-out participants do not need to have the same amount of available data as training participants to benefit from their decoder: with only 30 minutes of data – i.e. roughly 1000 samples – left-out participants can reach performance which would have required roughly 100 minutes of data – i.e 3000 samples – in a within-subject setting.

In addition, we show that a single decoder trained on all functionally aligned participants can reach better results than a decoder trained on all un-aligned participants (MR is 7.7 against 8.6 averaged across subjects), and performs on par with each corresponding single-subject decoders.

**Framework generality** Note that, in Figure 3, we chose the best performing participant (S2) as the reference participant. We report all other combinations of reference and left-out participants in Supplementary Figures S6 and S7 and find that all effects persist for all combinations. In addition, supplementary Figures S8 and S9 show that these results hold for all types of latents. Furthermore, we replicate this experiment on four participants from the Natural Scenes Dataset: decoding performance, reported in Tables S1 and S2, is higher than for the Wen 2017 dataset, probably due to using much more training data. Importantly, functional alignment allows the median rank to drop from 22.5 (baseline) to 11.5 on average in the out-of-subject setup for CLIP 257 × 768. In particular, it drops to 8.3 on average across left-out participants when S7 is used as reference. Finally, Figures S11 and S12 present all other setups we ran for this study. In particular, they show that a multi-subject aligned model (e.g. trained on S1 and S2) has better performance on aligned left-out participants (e.g. S3) than a single-subject model (e.g. trained on S2 only).

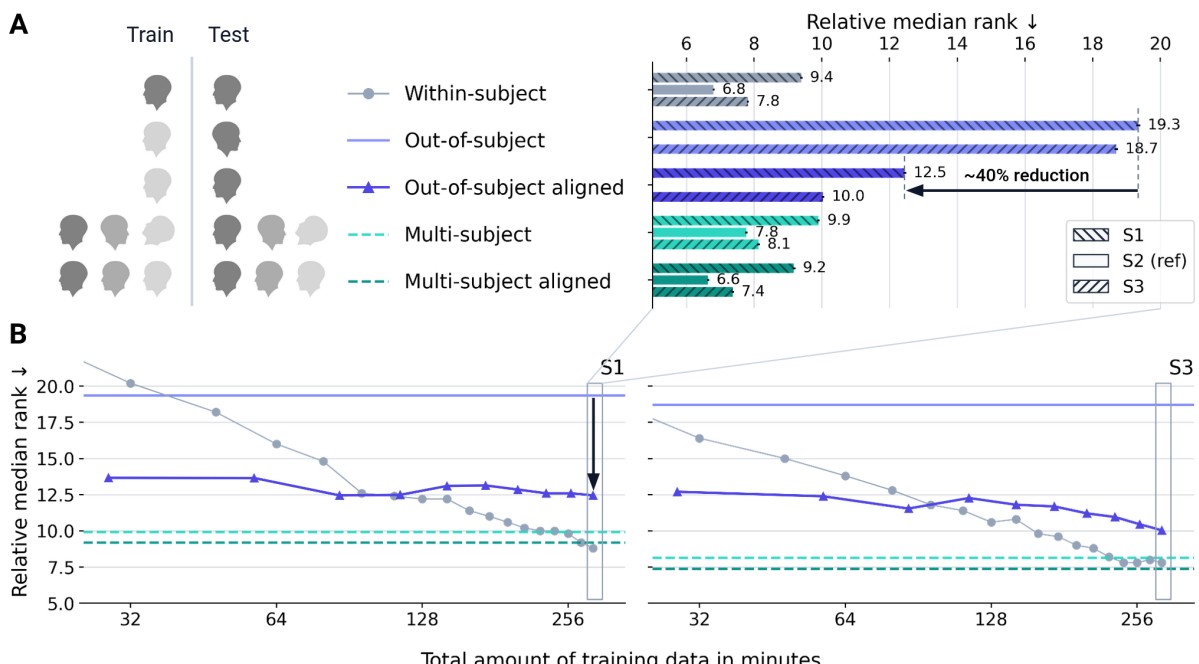

Figure 3: **Effects of functional alignment on multi-subject and out-of-subject setups**
We report relative median rank ↓ in all setups described in section 3.1 for CLIP $257 \times 768$. In all *aligned* cases, S1 and S3 were aligned onto S2. In all *out-of-subject* cases, we test S1 and S3 onto a decoder trained on S2. In all *multi-subject* cases, the decoder was trained on all data from all 3 participants. **A.** In this panel, all models (alignment and decoding) were trained on all available training data. Results for other latent types are available in Figure S8. **B.** In left-out S1 and S3, decoding performance is much better when using functional alignment to S2 (solid dark purple) than when using anatomical alignment only (solid pale purple). Performance increases slightly as the amount of data used to align participants grows, but does not always reach levels which can be achieved with a single-subject model fitted in left-out participants (solid pale gray dots) when a lot of training data is available. Training a model on multiple participants yields good performance in all 3 participants (dashed pale teal) which can be further improved by using functional alignment (dashed dark teal). Results for other latent types are available in Figure S9.

**Exploring computed inter-subject alignments**   To better understand how brain features are transformed by functional alignment, we show in Figure 4 how vertices from participant S1 are warped to fit those of participant S2. To this end, we colorize vertices in S1 using the MMP 1.0 atlas (Glasser et al., 2016) and use $\phi_{S1 \rightarrow S2}$ to transport each of the three RGB channels of this colorization to S2. Note that both participants' data lie on fsaverage5.
We see that, in low data regimes, FUGW does not recover a smooth inter-subject mapping of the cortical surface, but still manages to recover the cortical organization of the occipital lobe. A greater amount of data allows FUGW to reconstruct inter-subject mappings that are anatomically consistent in a much higher number of cortical areas such as the temporal and parietal lobes, and, unexpectedly, in the primary motor cortex as well. The prefrontal cortex and temporo-parietal junction (TPJ) seem challenging to map, perhaps due to greater inter-subject functional variability or lesser responsivity in those regions.

### 4.3   Influence of training set size and test set repetitions

Recent publications (Ozcelik & VanRullen, 2023; Scotti et al., 2023; Tang et al., 2023) in brain decoding using fMRI have shown impressive results, but these results are obtained using unusually large datasets and signal-to-noise ratios (e.g. tens of hours of 7T fMRI per participant). To evaluate the importance of these two factors, we report in Figure 5 performance metrics for models trained with various amounts of data and

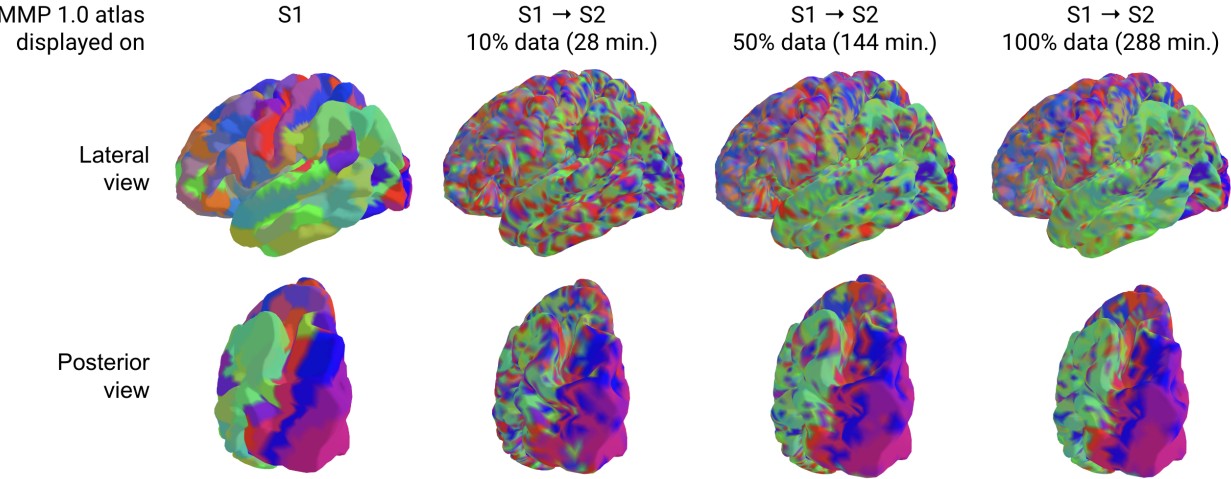

Figure 4: **Visualizing functional alignments in the left hemisphere** Vertices of the left-out participant (left column) are warped by FUGW. The result of this transport is visualized on the reference participant (columns 2, 3, and 4). Fitting FUGW with increasing amounts of data gradually leads the inter-subject mapping to better respect the cortical organisation of multiple areas, including non-visual ones. Note that all 3 models were fitted using the same number of iterations.

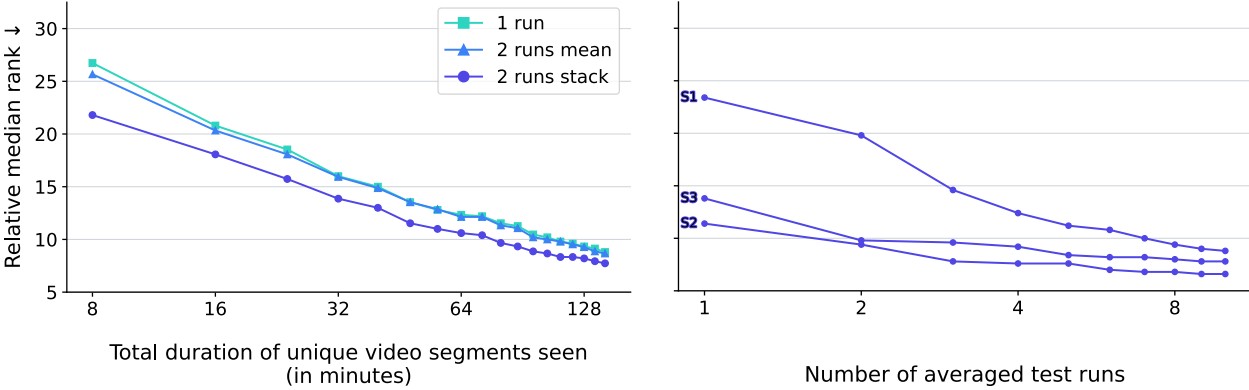

Figure 5: **Influence of training set size and test set noise** Relative median rank ↓ on a fixed test set averaged across participants gets better as more training data is used to fit the model (left). Interestingly, averaging brain volumes of 2 similar runs does not bring improvements compared to using just 1 run. Instead, stacking runs yields significant improvements. Note that training sets using 2 runs have twice as much data as those using 1 run. Finally, these metrics are highly affected by the noise level of the test set (right): averaging more runs in the test set yields better metrics despite using the same decoder.

tested with various amounts of noise.

Firstly, using a fixed test set, our results allow to systematically estimate the quantity of training data needed to achieve a given decoding performance. Interestingly, our results show that stacking two runs displaying the same stimuli yields better results than averaging them. Besides, for a given acquisition budget, showing different stimuli (as opposed to repeating stimuli) yields small but systematic performance improvements.

Secondly, reported performance metrics only hold in favorable signal-to-noise setups. Indeed, the test set associated with the Wen 2017 dataset comes with 10 runs for each video segment, which, when averaged together, greatly reduce the noise level. However, as reported in Figure 5, when tested in real-life signal-to-noise conditions (i.e. only one run per test video clip), our models' performance degrades: when using CLIP latents, for each participant, it is approximately twice as bad as when averaging all 10 runs.

## 5   Discussion

**Impact**   The present work confirms the feasibility of using fMRI signals in response to natural images and videos to decode high level visual features (Nishimoto et al., 2011). It further demonstrates that it is possible to leverage these fMRI signals to estimate meaningful functional alignments between participants, and use them to transfer semantic decoders to novel participants.

In particular, our study shows that decoding brain data from a left-out participant, i.e. a participant who was not used to train the decoder, can be substantially improved by aligning this left-out participant to a large reference dataset on which a decoder was trained. Our method thus paves the way to using models trained on large amounts of individual data to decode signals acquired in smaller neuroimaging studies, which typically record an hour or two of fMRI data for each participant (Madan, 2022). We also find that training a brain decoder on multiple functionally-aligned participants systematically improves decoding performance in these participants.

In addition, this study reports decoding accuracy in setups where participants are shown test stimuli for the first time, thus providing insight into how these models would perform in real-time decoding. While performance improves with the number of repetitions at test time, reasonable decoding performance of semantics can be achieved with only one repetition in two out of three participants.

Note that decoders can also be used to decode brain activity for which it is hard to provide a good latent representation, for instance when the participant is sleeping, or when dealing with animals. Our method could be used to decode brain activity in left-out participants in such cases, using a decoder that had previously been trained on a large amount of labelled data. Lastly, by systematically quantifying decoding accuracy as a function of the amount of training data, the present work brings insightful recommendations as to what stimuli should be played in future fMRI datasets collecting large amounts of data in a limited number of participants. In the current setup (naturalistic movie clips acquired at 3T), training with diverse semantic content is more valuable than training with repeated content for fitting decoding models.

**Limitations**   This work is a first step towards training accurate semantic decoders which generalize across individuals, but subsequent work remains necessary to ensure the generality of our findings.

Firstly, although the reported gains in out-of-subject setups are significant, the small number of participants present in the dataset under study requires replications on larger cohorts. However, to our knowledge, no other dataset has presented similar features to Wen et al. (2017), namely a large amount of data per participant and a large variety of video stimuli.

Secondly, our approach currently requires left-out participants to watch the same stimuli as reference participants. It is yet unclear whether functional alignment could bring improvements without this constraint. However, multi-subject decoding can probably help partially address this issue: since it is possible to train a decoder on multiple participants and because not all of them have to watch the same movies, it is possible that a lot of different movies - each seen by a different participant used in the training set - could be used as "anchors" for left-out individuals.

Finally, while restricting this study to linear models makes sense to establish baselines and ensure replicability, non-linear models have proved to perform as well (Scotti et al., 2023), and constitute a natural improvement of this work.

**Ethical implications**   Out-of-subject generalization is an important test for decoding models, but it raises legitimate concerns. In this regard, this study highlights that signal-to-noise ratio still currently makes it challenging to very accurately decode semantics in a real-time setup, and that a non-trivial amount of data is needed per individual for these models to work. In particular, it would be interesting to see if recent work in perception decoding in MEG (Défossez et al., 2022; Benchetrit et al., 2023) could be applied to out-of-subject setups with a method similar to ours. Moreover, we stress that, while great progress has been made in decoding perceived stimuli, imagined stimuli are still very challenging to decode (Horikawa & Kamitani, 2017). Nonetheless, it is important for advances in this domain to be publicly documented. We thus advocate that open and peer-reviewed research is the best way forward to safely explore the implications of inter-individual modeling, and more generally brain decoding.

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

# A   Appendix

## A.1   Data pre-processing

We strive to minimally preprocess acquired BOLD signal. To this end, we detrend acquired BOLD signal (i.e. we remove cosine drifts) and finally standardize voxels' timecourses for each run, as shown in Figure S1.

Moreover, when decoding the latent representation of a given image, we use brain volumes which have been acquired after the image's onset. Figure S2 illustrates this idea, and introduces the concepts of *window size* (i.e. the number of brain volumes we use) and *lag* (i.e. the time difference between the onset of the image to be decoded and the first brain volume used to decode it). Values for both of these hyper-parameters were obtained through a 5-fold cross-validated grid search over samples of the training set. We report these results in Figure S3.

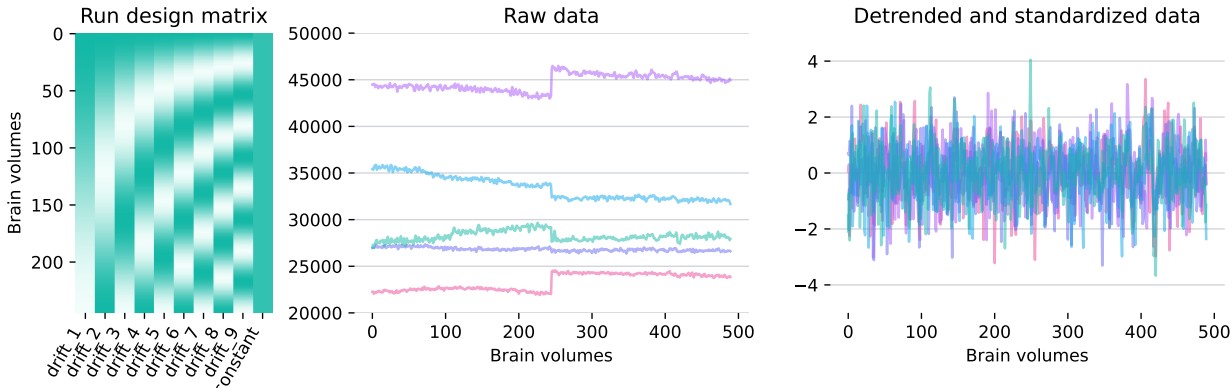

Figure S1: **Pre-processing of the Wen 2017 dataset** For each participant and each run, in each vertex, we regress out parts of the signal which can be linearly explained by the design matrix represented on the left, which models cosine drifts of the BOLD signal. The two graphs to the right show time-courses in 5 vertices across 2 different runs before (left) and after (right) they have been pre-processed.

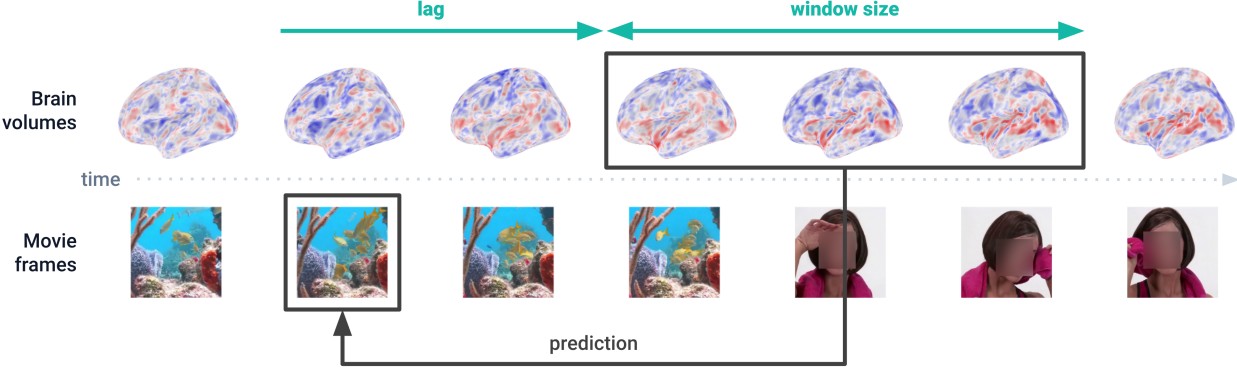

Figure S2: **Lag and window size** In order to decode a movie frame which was seen at time $t$, one can use brain volumes which were acquired further in time. This delay is referred to as the *lag*. Moreover, one can use several brain volumes to decode a given movie frame. The number of brain volumes used is called the *window size*. Images featuring human faces were blurred.

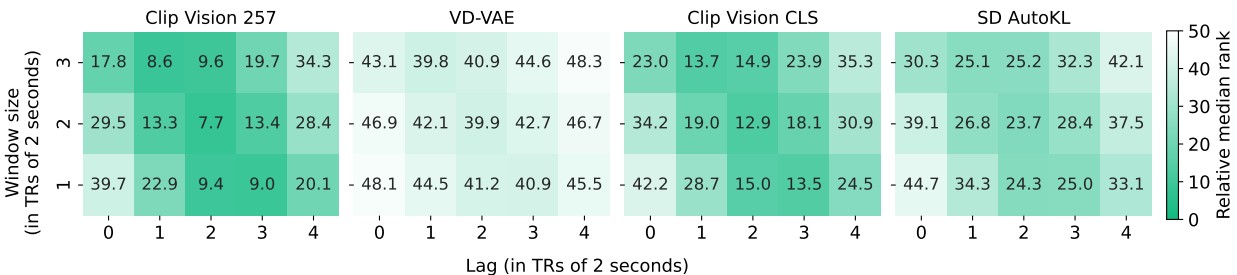

Figure S3: **Relative median rank ↓ of predicted latents averaged across participants for various time lags and window sizes**

### A.2 Retrieving images using predicted latent representations

Predicted latent representations can be compared to that of images in a retrieval set. In Figure S4, for each image shown to the participant during the test phase, we print the five images from the retrieval set whose latent representation is the closest to predicted latents. We see that semantics are often preserved.

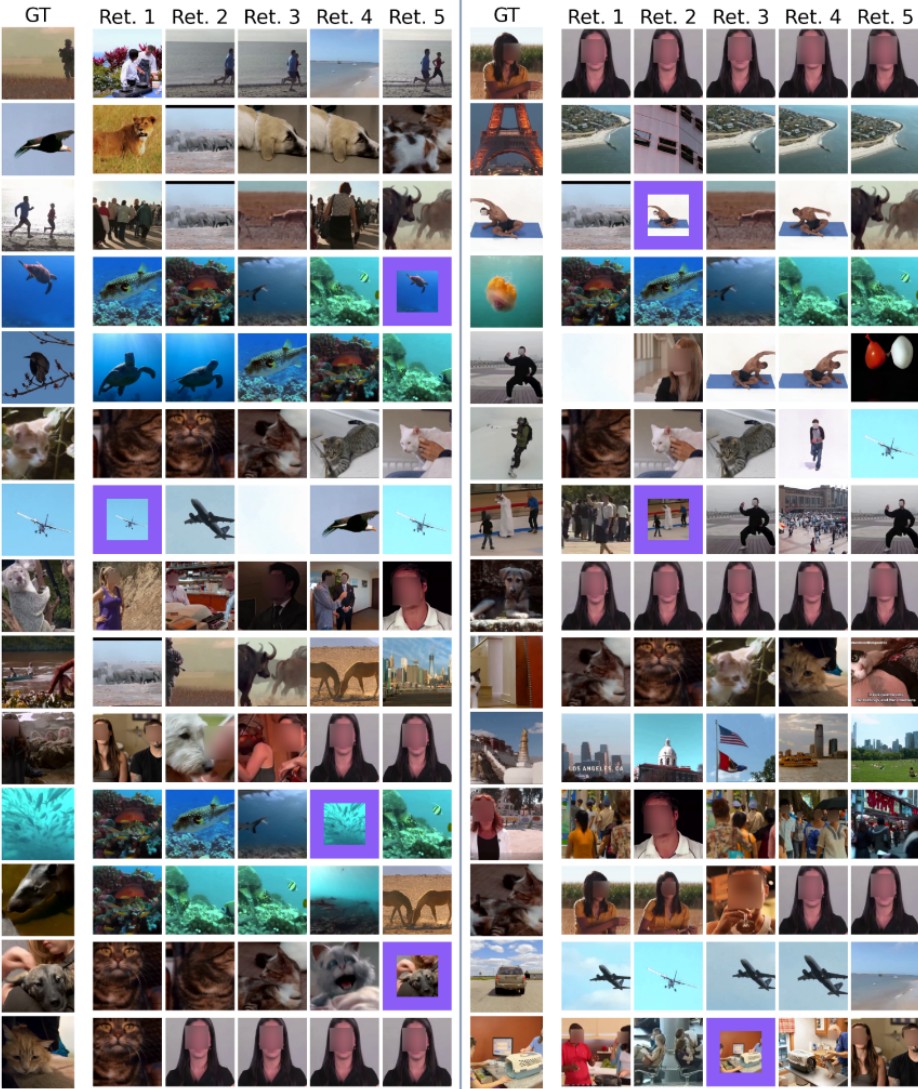

Figure S4: **Image retrievals using predicted latent representations of CLIP 257 × 768 latents** We use a model fitted on Subject 2 (S2) from the Wen 2017 dataset and predict the latent representation of unseen videos (test set). Ground truth (GT) images featured within the first 5 retrieved (Ret.) images are indicated with a bold purple border. In a given row, images which appear similar across columns are actually different frames of the same video clip. Images featuring human faces were blurred.

## A.3 Results for every combination of reference participant and left-out participant

Figure S5 is a copy of Figure 3 from the main pages of this paper. It illustrates the main effects reported in our study, namely that (1) functional alignment yields better performance than anatomical alignment when transferring a semantic decoder to left-out individuals, (2) it is possible to train such decoders on multiple participants and (3) this last setup works best when participants are aligned.

Figure S5 only shows these results when participant 2 of the Wen 2017 dataset is used as the reference participant. Therefore, we add Figures S6 and S7, which illustrate that all results hold regardless of what participant of the cohort is used as reference or left-out participant.

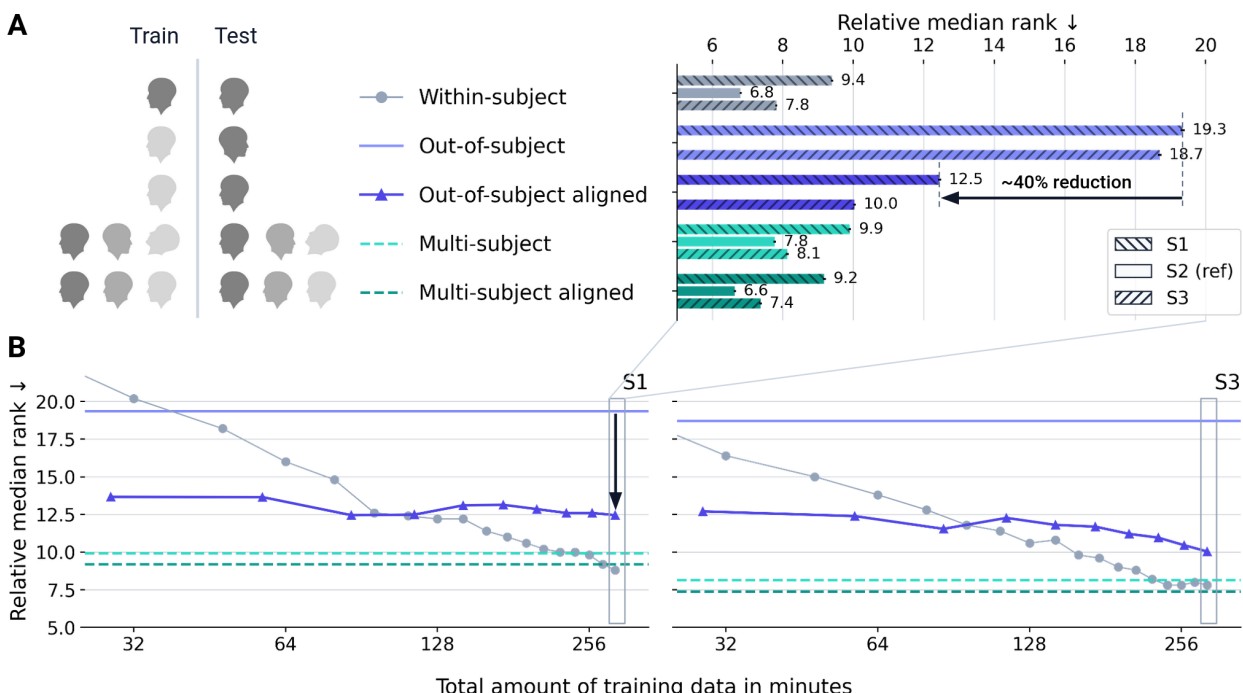

Figure S5: **Effects of functional alignment on multi-subject and out-of-subject setups using participant 2 as the reference participant** We report relative median rank ↓ in all setups described in section 3.1 for CLIP $257 \times 768$. In all *aligned* cases, S1 and S3 were aligned onto S2. In all *out-of-subject* cases, we test S1 and S3 onto a decoder trained on S2. In all *multi-subject* cases, the decoder was trained on all data from all 3 participants. **A.** In this panel, all models (alignment and decoding) were trained on all available training data. Results for other latent types are available in Figure S8. **B.** In left-out S1 and S3, decoding performance is much better when using functional alignment to S2 (solid dark purple) than when using anatomical alignment only (solid pale purple). Performance increases slightly as the amount of data used to align participants grows, but does not always reach levels that can be achieved with a single-participant model fitted in left-out participants (solid pale gray dots) when a lot of training data is available. Training a model on multiple participants yields good performance in all 3 participants (dashed pale teal) which can be further improved by using functional alignment (dashed dark teal). Results for other latent types are available in Figure S9.

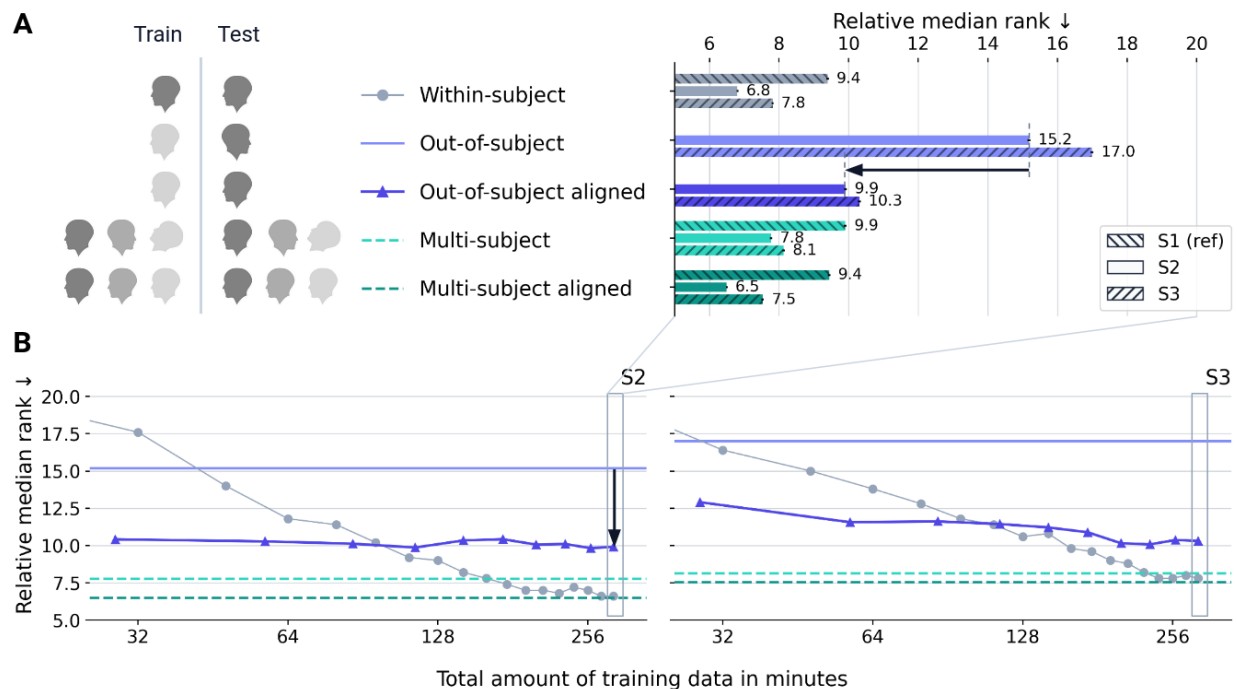

Figure S6: **Effects of functional alignment on multi-subject and out-of-subject setups using participant 1 as the reference participant**

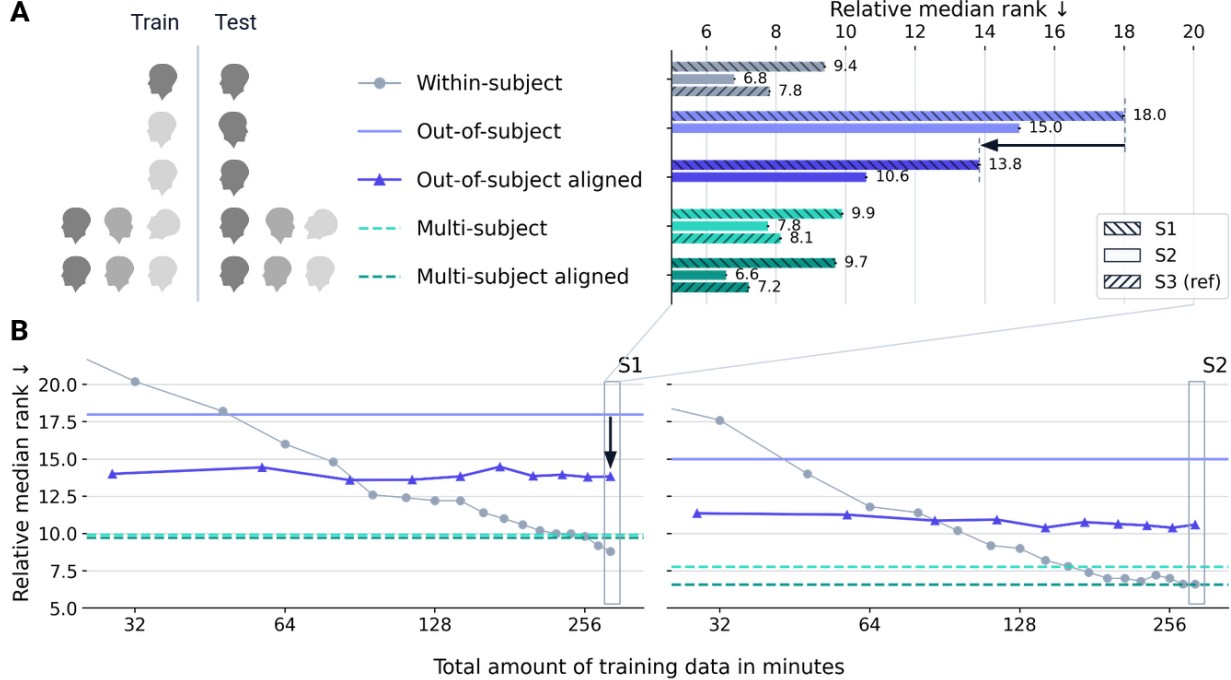

Figure S7: **Effects of functional alignment on multi-subject and out-of-subject setups using participant 3 as the reference participant**

## A.4 Results for every type of latent representation

In this section, we extend the claims made in Figures 3.A, 3.B and 5 by showing that these results hold for other latent representations, namely VD-VAE, CLIP CLS and AutoKL. Figures S8, S9 and S10 extend Figures 3.A, 3.B and 5 respectively, showing that observed effects are present regardless of the chosen latent representation. All of these figures were obtained using the Wen 2017 dataset.

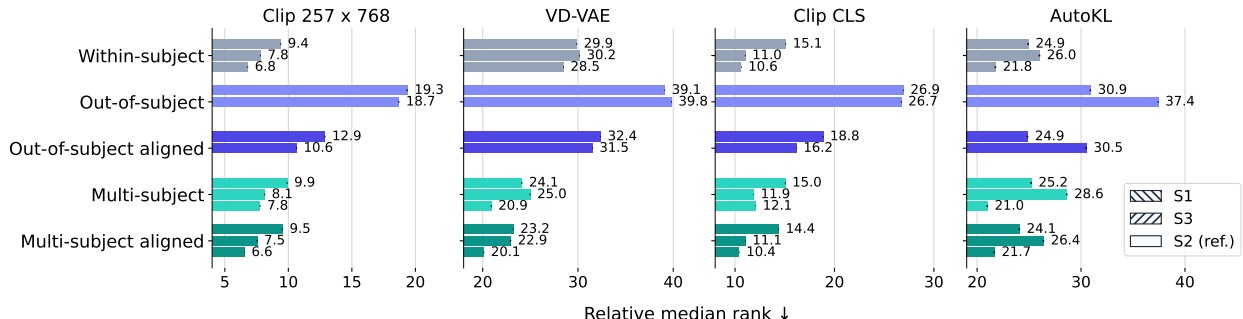

Figure S8: **Effects of alignment** For any type of latent representation, out-of-subject decoding performance, measured through relative median rank ↓, greatly improves when participants are functionally aligned. Training decoders on multiple participants also works better when participants are aligned. These results were averaged across 50 retrieval sets ; all these metrics are reported with a standard error of the mean (SEM) smaller than 0.01.

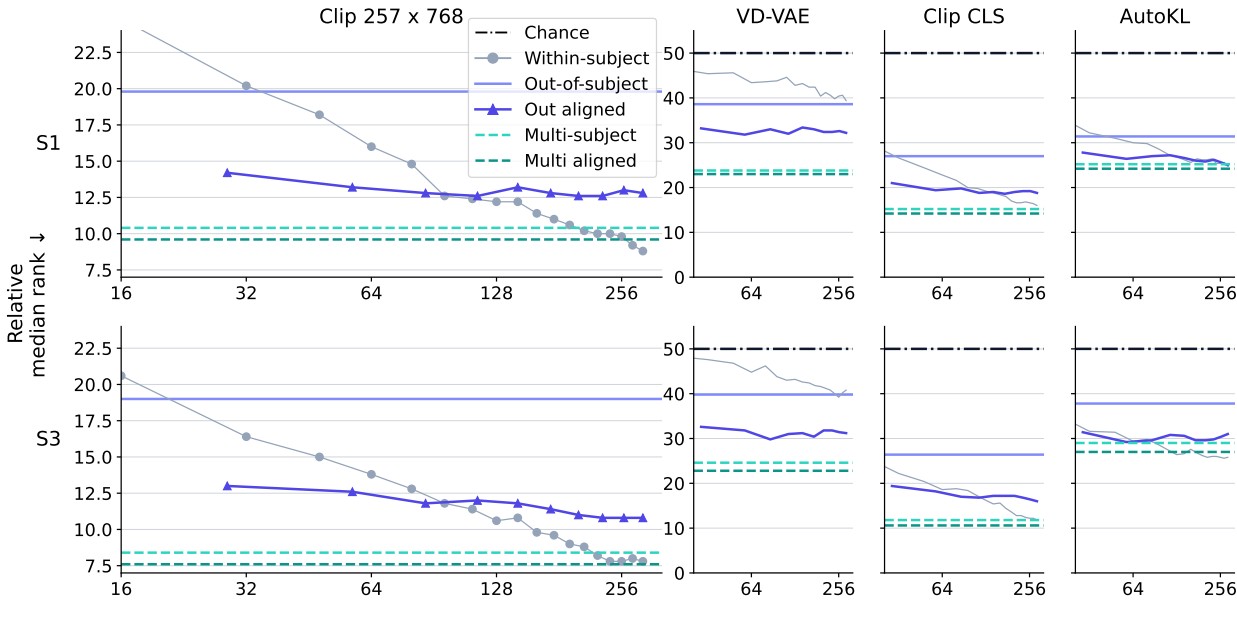

Figure S9: **Performance increases slightly with more alignment data** For any type of latent representation, out-of-subject decoding performance greatly increases with functional alignment even in low data regimes. In high data regimes, out-of-subject decoding does not work as well as fitting single-subject or multi-subject models.

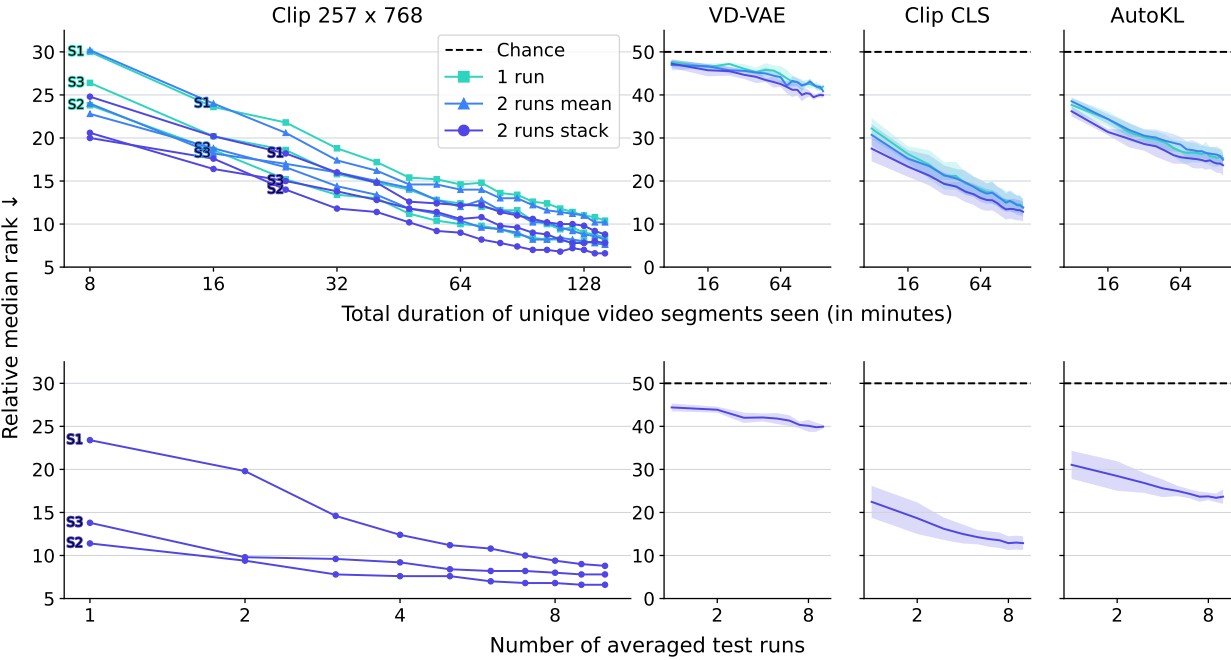

Figure S10: **Scaling studies for all latents** For any type of latent representation, decoding performance increases linearly with exponentially more data. It also seems that, when acquiring data at 3T or more, not repeating stimuli yields the best results. At test time, although repeating stimuli allows to get better metrics, retrieval performance with only one repetition is already reasonable in 2 out of 3 participants of the Wen 2017 dataset.

### A.5 Decoding results for all setups

On top of experiments reported in the main pages of this paper, we have tested a lot of different training and tests sets. In this section, we report the Relative Median Rank ↓ for all 97 training sets and all 63 test sets, and CLIP latent representations. Training sets include all possible single-subject, multi-subject unaligned and multi-subject aligned cases. Test sets include all possible with-subject, left-out unaligned and left-out aligned cases. Every time, all available training sessions are used for training the decoder. However, we vary the amount of data used to train the alignments, for both training and test sets.

We report detailed results for CLIP $257 \times 768$ and CLIP CLS in Figures S11 and S12 respectively.

In particular, these figures report combinations our setups of interest which were not mentioned in the main text. We find of particular interest the multi-subject out-of-subject setup, in which the decoder has been trained on two (un)aligned participants and tested on a third (un)aligned one.

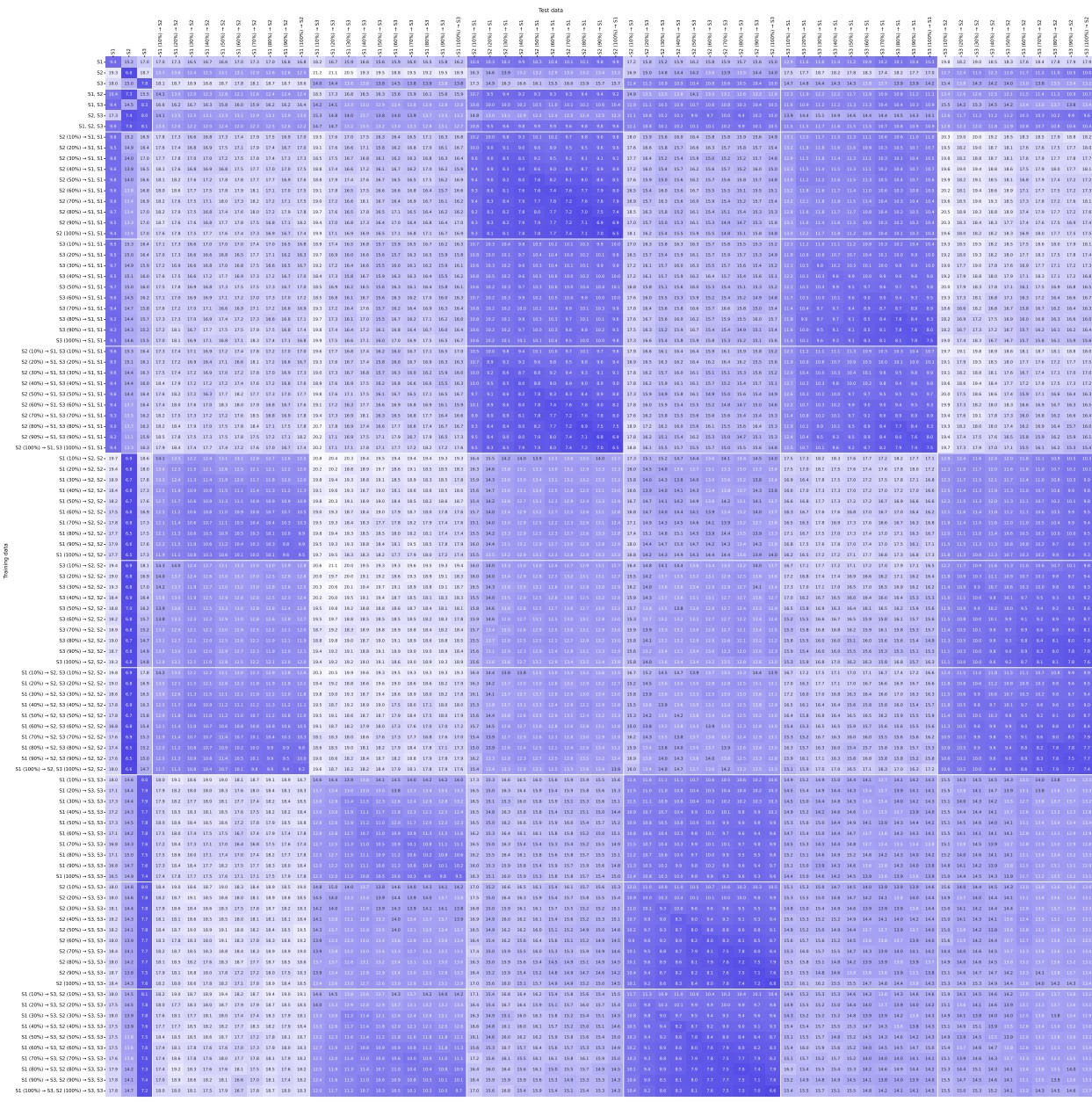

Figure S11: Relative median rank ↓ for **CLIP 257 × 768** latents in single- and multi-subject training sets, with and without alignment, tested on within- and across-participants setups with and without alignment. These results were averaged across 50 retrieval sets ; all these metrics are reported with a standard error of the mean (SEM) smaller than 0.01.

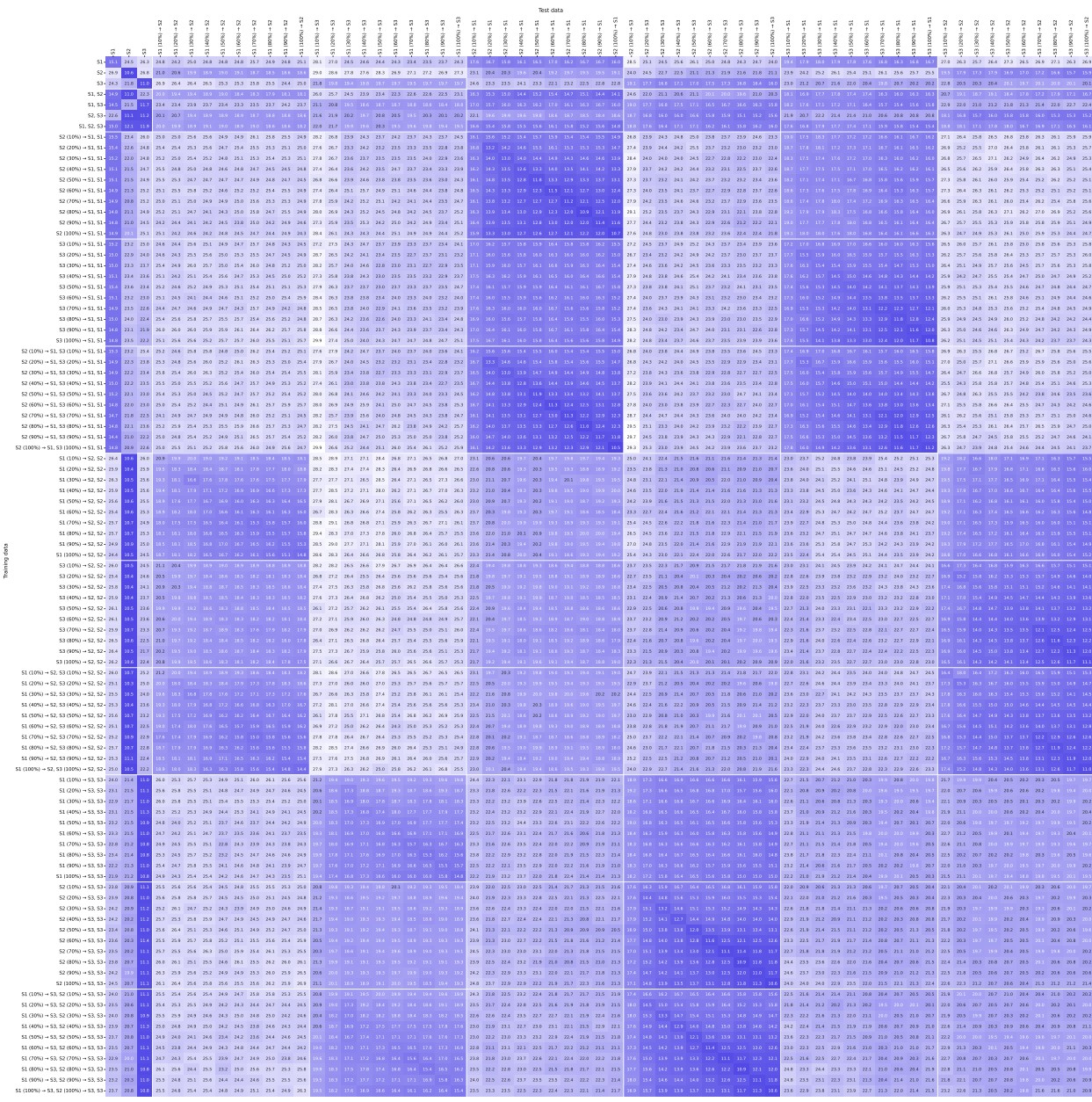

Figure S12: Relative median rank ↓ for **CLIP CLS** latents in single- and multi-subject training sets, with and without alignment, tested on within- and across-participants setups with and without alignment. These results were averaged across 50 retrieval sets ; all these metrics are reported with a standard error of the mean (SEM) smaller than 0.01.

### A.6   Replication on the Natural Scenes Dataset

We replicate our main experiment using data from the Natural Scenes Dataset (Allen et al., 2022). This dataset comprises 8 participants who each see 10 000 images 3 times, thus leading to a total of 30 000 trials per participant. For each participant, this data is acquired in 40 sessions of 60 minutes each. For each participant, there is a total of 1 000 images which are shared with other individuals - i.e. other individuals will see them too - and 9 000 which are exclusive - i.e. other individuals will not see them. We sub-selected all participants who had completed all 30 000 trials, namely participants 1, 2, 5, and 7. For each selected participant, we split their 30 000 trials in two sets: all exclusive images are grouped in the *decoding set* and all shared images are grouped in the *alignment set*. We further split the decoding set into disjoint sets of images for training and testing individual decoders, whose performance is reported in Table S1. Alignments sets are used to compute functional alignments between individuals. Besides, we used pre-computed beta coefficients computed with GLM denoise on *fsaverage7* (Fischl, 2012) and openly available online. We down-sampled this data to *fsaverage5* - which simply amounts to keeping only the first 10 242 array elements in each hemisphere.

Eventually, we show that decoders tested on left-out individuals work consistently and significantly better when left-out participants are functionally aligned rather than simply anatomically aligned to the reference participant, as reported in Table S2.

Table S1:   **Within-subject metrics for all NSD participants and all latent types on the test set** Reported metrics are relative median rank ↓ (MR) of retrieval on a set of 500 samples, top-5 accuracy % ↑ (Acc) of retrieval on a set of 500 samples. Chance level is at 50.0 and 1.0 for these metrics respectively. These results were averaged across 50 retrieval sets, hence results are reported with a standard error of the mean (SEM) smaller than 0.01.

|    | CLIP 257 × 768 | | VD-VAE | | CLIP CLS | | AutoKL | |
|----|------|------|------|-----|------|------|------|-----|
|    | MR   | Acc  | MR   | Acc | MR   | Acc  | MR   | Acc |
| S1 | 3.6  | 26.6 | 23.0 | 4.6 | 4.6  | 19.1 | 30.5 | 1.8 |
| S2 | 6.0  | 17.6 | 22.1 | 4.4 | 6.9  | 13.8 | 33.6 | 1.5 |
| S5 | 4.6  | 19.9 | 26.0 | 3.7 | 4.3  | 19.7 | 31.5 | 2.5 |
| S7 | 4.0  | 24.4 | 23.4 | 5.4 | 5.5  | 18.5 | 24.5 | 4.3 |

Table S2: **Across-subject metrics for all NSD participants and all latent types on the test set** We report the decoding performance of decoders trained on a reference participant and tested on a left-out participant who was anatomically aligned (A) or functionally aligned (A+F). The reported metric is the relative median rank ↓ (MR) of retrieval on a set of 500 samples. These results were averaged across 50 retrieval sets, hence results are reported with a standard error of the mean (SEM) smaller than 0.01. One sees that functionally aligned data is always better decoded than anatomically aligned data. In particular, when S7 as the reference subject, functional alignment helps divide the median rank by 3 for CLIP latents.

| Reference | Left-out | CLIP 257 × 768 | | VD-VAE | | CLIP CLS | | AutoKL | |
|---|---|---|---|---|---|---|---|---|---|
| | | A | A+F | A | A+F | A | A+F | A | A+F |
| | S2 | 20.9 | **10.7** | 35.2 | **24.5** | 28.3 | **13.4** | 42.5 | **37.1** |
| S1 | S5 | 32.2 | **13.6** | 43.1 | **32.0** | 33.4 | **12.5** | 43.9 | **37.7** |
| | S7 | 33.5 | **14.7** | 40.8 | **30.7** | 36.1 | **17.1** | 45.0 | **37.0** |
| | S1 | 18.3 | **10.4** | 37.0 | **30.4** | 24.4 | **14.4** | 35.9 | **37.1** |
| S2 | S5 | 29.3 | **12.0** | 40.3 | **34.1** | 30.8 | **11.2** | 42.4 | **39.6** |
| | S7 | 27.9 | **14.6** | 40.2 | **32.8** | 32.1 | **17.9** | 41.8 | **38.9** |
| | S1 | 29.2 | **13.0** | 43.4 | **34.4** | 33.8 | **14.4** | 37.8 | **33.8** |
| S5 | S2 | 26.2 | **9.8** | 40.1 | **30.1** | 31.4 | **11.2** | 36.0 | **36.2** |
| | S7 | 29.8 | **14.1** | 41.7 | **32.0** | 34.4 | **17.3** | 40.0 | **34.9** |
| | S1 | 27.7 | **7.7** | 43.1 | **30.1** | 32.3 | **11.5** | 40.6 | **26.5** |
| S7 | S2 | 23.4 | **8.6** | 38.9 | **26.4** | 28.6 | **13.2** | 37.8 | **29.1** |
| | S5 | 27.5 | **8.8** | 41.6 | **31.2** | 30.4 | **9.8** | 43.4 | **30.9** |

