# OpenReview forum: "Sample-efficient decoding of visual stimuli from fMRI through inter-individual functional alignment"
_TMLR — Accepted by TMLR_

### Review · Reviewer_Ba96 · 2024-06-21

**Summary Of Contributions:**

The paper introduces a new system to decode fMRI scans of subjects into latent variables. The primary benefit of this model is that it generalizes across subjects via an alignment model. This is a way of getting more data and introducing noise which makes the model more robust and produce stronger latents. These latents were evaluated using a retrieval task.

**Audience:**

Yes

**Broader Impact Concerns:**

It could be a serious privacy violation to decode ones thoughts via fMRI.

**Claims And Evidence:**

Yes

**Requested Changes:**

More ablations could be done around the loss terms or different encoders.

I would be curious to see another evaluation task. Perhaps zero shot text captioning from the CLIP embeddings? Or decoding images from the VAE?

**Strengths And Weaknesses:**

The goals of the paper are focused, and the approach is clear and intuitive. In some ways, this paper is not unlike machine translation, where each subject speaks its own language. The paper indeed demonstrates feasibility of using multiple subjects to improve an fMRI decoder.

The authors are pretty clear about the limitations and some weaknesses. The data requirements are heavy. Each subject has to watch the same videos and a separate alignment model.

Evaluation could be better. Retrieval requires a database of existing latents, and it would be good to have some type of downstream task that only uses the latents.

---

> ### Author Response · Authors · 2024-10-26
> **Response from the authors**
>
> We thank the reviewer for their time and input!
>
> We address your comments point by point:
> - given the high number of setups we tested in this study, we strive to limit the number of varying parameters, and therefore refrained from testing too many different losses or encoders. Note that other papers have focused on the point you raise here, however they had to narrow their study to within-subject setups: Benchetrit, Yohann, Hubert Banville, and Jean-Rémi King. ‘Brain Decoding: Toward Real-Time Reconstruction of Visual Perception’. arXiv, 14 March 2024. [http://arxiv.org/abs/2310.19812](http://arxiv.org/abs/2310.19812).
> - similarly, we refrain from testing downstream tasks because (1) we feel that the different within- and across-subject setups are already a bit complex to grasp and (2) we advocate that evaluating metrics on downstream tasks using pre-trained generative models is hard to interpret. Indeed, in case the downstream task metrics are poor, it is hard to determine the respective influence of brain decoder and the downstream generative model on these bad results. Instead, we focus on a retrieval task because it can directly assess the performance of the brain decoder.
>
> We hope to have addressed the reviewer's concern and are open to discussing more changes if need be!

---

### Review · Reviewer_qt2z · 2024-06-28

**Summary Of Contributions:**

The paper deals with the task of mapping brain signals to neural network representations. In particular, the brain signals are a dataset of $X \in R^{N \times V}$ and the target neural network representations $Y \in R^{N \times M}$, where $N$, $V$ and $M$ are the number of brain scans, number of voxels and representation size. A linear mapping from brain signals to NN representations is learned using ridge regression.

The paper mainly focuses on the "out-of-subject" setup where the decoder is learned on one subject and then applied/transferred to 1 or more “out-of-distribution” subjects. They rely on (Thual et al 2022) to learn an alignment transformation between brain signals of the out-of-subject $X \in R^{M \times V}$ and in-subject $X \in R^{N \times V}$. One can then apply the same “in-subject” decoder to predict NN representations.

Results are shown on Wen et al 2017 in the main paper and Allen et al 2021 (Natural Scenes Dataset) in the supplementary. The authors initially validate the linear decoder after reporting in-distribution metrics for three subjects S1, S2 and S3. They choose S2 as the in-distribution subject and study the transfer on S1 and S3. The main contributions are:

* The alignment approach is better than learning a separate decoder directly on out-of-subject in the limited data regime.
* Their approach is better than applying the "within-subject" decoder on "out-of-subject."
* Further, they show that training a single decoder on aligned data, works better than training a single decoder on unaligned data.

**Audience:**

Yes

**Claims And Evidence:**

Yes

**Requested Changes:**

* Some changes can be made to the structure of the paper. The paper combines the background work (Section 2.1 and Section 2.2) with their contributions (Section 2.3). It is better to separate this, so the readers can understand the exact contribution quickly.
* The experimental setup (Section 2.4 - Section 2.6) is also merged with the methods section. The authors can move the experimental setup to a new section.
* The functional alignment seems extremely useful, in cases where the “left-out” subjects do not have target latent representations. $Y \in R^{n \times m}$. The authors can consider adding some discussion about this to the main text.
* There may be some potential overclaims. “Compared to the baseline, functional alignment across participants boosts visual semantics decoding performance in left-out participants, especially when the latter have a limited amount of data”. From the paper’s results, it does not boosts decoding performance in left-out participants when there is enough data. For example, In Fig 3. A, S1 and S3 achieve within-subject performance of 9.4 and 7.8 as compared to 12.5 and 10.0 with functional alignment. So if one has enough data, one can directly train decoders on 'out-of-subject" data. Please fix these in the introduction and other places.
* What is exactly v in the functional alignment setting? Is it still 10242?
* The paper converts cosine similarity to a retrieval-based rank. Why not report mean cosine similarity between the predicted and target clip representations directly?
* What metric is used to tune the hyperparameters in the cross-validated grid search?
* In various sections of the paper, the authors use the minutes notation, which may be standard in neuroscience. It is also helpful to convert this to a number of examples. So that, general readers from the machine learning community can get a sense of the scale of the data.

Not Critical:
* Fig 5 is crowded. Just displaying S1 may be sufficient.
* Replace Model A and Model B with low-level and high-level decoder.
* Both CLIP and VD-VAE can retain both high-level and low-level representations of the input. So I'm not 100% sure of the low-level/high-level demarcation.

**Strengths And Weaknesses:**

The paper is easy to read and the authors provide a technique to boost out-of-subject performance in the limited data regime. In general, the claims seem validated. However I defer to the AC/other reviewers on the generality of the datasets used in the paper as I am not an expert on fMRI data.

There are a few improvements that can be made in presentation of the paper or claims. See below for a list of requested changes, the authors can consider updating the draft in response to the requested changes.

---

> ### Author Response · Authors · 2024-10-26
> **Response from the authors**
>
> We thank the reviewer for their time and questions!
>
> We address the requested changes point by point below.
> - as suggested, we separated our methodological contributions to make things clearer
> - as suggested, we created a new section presenting our experimental setup
> - cases where the left-out subjects do not have target latent representations could in theory be addressed with our method, although left-out subjects would still need some labelled data in order to be functionally aligned to within subjects. We touch upon this at the end of the paper
> - indeed, the formulation feels ambiguous ; we changed it to better represent our results
> - yes, it is 10242 ; we changed the text to make that clear
> - we report retrieval-based metrics because (1) they are tightly connected to the loss our decoders are trained on and (2) we find them more informative than simple cosine similarity. We could report cosine similarity distributions, which would in the end be comparable to reporting rank distributions. Other papers have tried using mixed losses combining a retrieval loss and cosine similarity, which we refrain from doing here for simplicity
> - the metric used for cross-validation is the relative median rank introduced in section 2.2. We updated the text to make this clear.
> - this is indeed a valuable piece of information. We added sample sizes throughout the document
>
> Not critical points:
> - we fused curves across participants to make the figure less crowded: https://ibb.co/M9tnFyb
> - as suggested, we replaced the labels to make the figure clearer
> - indeed, this is a fair point. This distinction was emphasized in Scotti et al. 2023, and we went on with the same distinction in our paper
>
> We hope to have addressed the reviewer's concern and are open to discussing more changes if need be!

---

### Review · Reviewer_bDCE · 2024-11-18

**Summary Of Contributions:**

1. The proposed functional alignment of the brain feafures across participants boosts visual semantics decoding performance in left-out participants, which have a limited amount of data compare to reference participants.

2. Evaluated on a retrieval task, compared to the anatomically-aligned baseline, the paper halves the median rank in out-of-subject setups.

3. The paper show this method aligns neural representations in accordance with brain anatomy.

**Audience:**

Yes

**Broader Impact Concerns:**

Once the problem is perfectly solved in the future, can we read what people see from their fMRI? Will this cause privacy issues? Furthermore, visual signals can also be changed into auditory signals, tactile signals, and taste signals. So can we recover the external signals that people are exposed to through fMRI or other physiological data?

**Claims And Evidence:**

Yes

**Requested Changes:**

1. I suggest replacing the brain decoders in the figure annotation in figure 1 with model A and model B to improve readability.
2. The functional data in 2.1 Functional alignment  need to be more detailed description to improve readability.
3. Out-of-subject is hard to defined, How do the work define what kind of participants are reference participants and what kind of participants are left-out participants? Is it based on artificial engineering experience? Or is there a physiological basis? Please explain in the paper.
4. This paper should analyze the impact of the ratio of Left-out and referecen-out on the alignment model, and quantify this impact into the final evaluation index.

**Strengths And Weaknesses:**

1. Strengths:

1.1 The paper basically describes how the proposed method works.

1.2 The paper claims that it can make models trained on large amounts of data work on certain "outliers points". This paper focuses on good problems.

2. Weaknesses:

2.1 The predicted latent representation does not seem to be very accurate. In fact, from Figure 2, I see that by retrieving the images with the latent representation that is closest to the predicted latent representation. The image seen by the participant can't be recovered(predicted). Especially the Eiffel Tower prediction. Also, from the Figure S4 of appendix, the predicted latent representation of dog seem to be to recovered to a human.

2.2 The overall framework has practical problems, assuming that in the future work this type of overall technical route can obtain a very accurate predicted latent representation, in other words, even though the problem in 2.1 is perfectly solved by other work in the future(but the problem don't be solved in this paper), the framework can only guess what a person has seen through retrieval. Once what a person has seen is not in the database, then logically speaking, it is impossible to retrieve what the person has seen. If it is impossible to guess what humans actually see due to the content of the database, what is the meaning of predicting latent representation? If it is meaningful, please provide specific experimental scenarios and experimental data to illustrate the usage of the predicted latent representation.  I can give a solution as a reference (if it is wrong, please point out my problem), for example, using the generative model to amplify the latent variables decoded from brain features, and then map the amplified new latent variables to the image space, which can avoid the problem of missing content in the database.

2.3 Out-of-subject is hard to defined, How do the work define what kind of participants are reference participants and what kind of participants are left-out participants? Is it based on artificial engineering experience? Or is there a physiological basis? Please explain in the paper.

2.4 In fact, even if the problem in 2.3 is perfectly defined, the alignment model will not work when the number of left-out participants is small enough. So there is a problem here. This paper should analyze the impact of the ratio of Left-out and referecen-out on the alignment model, and quantify this impact into the final evaluation index.

2.5 The encoding and decoding model part and the alignment model part of this paper are trained separately, so there is error accumulation, which will make the work not robust. The 3 human participants in the first dataset and 8 human participants in the second dataset should be expanded to prove that the method's robustness.

---

> ### Author Response · Authors · 2024-11-29
> **Response from the authors**
>
> We thank the reviewer for their time and input!
>
> First, we respond to the points they raised one by one:
> - indeed, decoded latent representations could very likely be improved by collecting a larger training dataset. Note that the dataset used in this study is much smaller than that of other papers tackling brain decoding with fMRI (2 hours here vs 10+ hours for [1] and [2]). We modify our discussion paragraph to emphasise this point
> - as pointed-out by the reviewer, predicted latent representations can be used as inputs for downstream generative models (see [3] and [4]), which alleviates the problem of not finding the decoded stimuli in the retrieval dataset. We refrain from implementing this part in our paper, because these downstream tasks' evaluation metrics depend on the chosen generative model. Instead, we focus on retrieval metrics exclusively
> - the out-of-subject setup is defined in section 3.1: out-of-subject means that data used to test the decoder was acquired in participants who were not used to train the decoder. We refer to these participants as "left-out participants". It is the opposite of the within-subject setup, in which test participants are the same as training participants. Moreover, any participant who was part of the training dataset can be used as a reference participant, i.e. a participant used for functional alignment (as described in section 2.1). Note that, in this paper, we tested all possible combinations of left-out / reference participants.
> - in principle, any participant who is not part of the training dataset can be functionally aligned to one of the reference participants, provided that they have seen the same stimuli
> - actually, we choose to freeze the encoder in order to enable downstream generative tasks using pre-trained models, as suggested by the reviewer in their previous point
>
> Then, we respond to the requested changes one by one:
> - reviewer qt2z suggested a similar change, which we implemented
> - functional data used for alignment are similar to that used for training / testing decoders. We swap sections 2.1 and 2.2 to improve readability. Moreover, supplementary materials describe these data and their pre-processing in more detail
> - see answer above ; we modify the discussion paragraph to stress that left-out participants data has not been used to train the brain decoder
> - see answer above ; we modify the discussion paragraph to stress that our decoders perform better in out-of-subject setups when trained on several participants
>
> We hope to have addressed the reviewer's concerns and are open to discussing more changes if need be!
>
> [1] Tang et al., ‘Semantic Reconstruction of Continuous Language from Non-Invasive Brain Recordings’. 2022
>
> [2] Scotti et al., ‘Reconstructing the Mind's Eye: fMRI-to-Image with Contrastive Learning and Diffusion Priors’. 2023
>
> [3] Ozcelik and VanRullen, ‘Natural Scene Reconstruction from fMRI Signals Using Generative Latent Diffusion’. 2023
>
> [4] Benchetrit, Banville, and King, ‘Brain decoding: toward real-time reconstruction of visual perception’. 2024

---

### Decision · Action_Editor_Dy5a · 2025-04-02

**Recommendation:** Accept with minor revision

**Comment:**

The strengths of this paper include
1) The proposed functional alignment method is novel and well-motivated.
2) The quantitative results are solid, showing superior decoding accuracy compared to baseline models.
3) The paper is well-written and clear, making complex ideas accessible.

This paper has been reviewed by two experts: one recommends acceptance, while the other leans toward rejection. The Associate Editor (AE), who is also an expert in this field, has evaluated the concerns raised.
Although one reviewer remains concerned that the latent representations may not be learned effectively enough, the task addressed in this paper is inherently complex and poses challenges that likely require broader community efforts beyond the scope of this work.

Overall, the paper supports its claims with empirical results and holds potential interest for the neuroscience community. It is well-structured, self-contained, and in good shape for acceptance. The authors should be encouraged to improve the camera-ready version by incorporating the reviewers' comments.

**Audience:**

The paper is primarily targeted at neuroscientists, machine learning researchers, and cognitive scientists working on fMRI decoding.
It is also relevant to brain-computer interface (BCI) researchers and those exploring multi-subject learning for neural decoding.
Recently, the researchers from computer vision and machine learning communities are also interested in these topics.

**Claims And Evidence:**

The paper presents a sample-efficient method for decoding visual stimuli from fMRI signals through inter-individual functional alignment.
It claims to improve generalization across subjects and reduce training data requirements.
The approach is supported by quantitative evaluations, demonstrating better performance than traditional fMRI decoding methods.
The evidence includes benchmark comparisons and visual reconstructions, validating the alignment method.

---

> ### Author Response · Authors · 2025-04-17
> **Camera ready version**
>
> We thank the action editor for their positive response and feedback!
>
> We have uploaded a camera-ready version incorporating the reviewers' comments.